# Public Health Services, Health Human Capital, and Relative Poverty of Rural Families

**DOI:** 10.3390/ijerph191711089

**Published:** 2022-09-04

**Authors:** Yingya Yang, Liangliang Zhou, Chongmei Zhang, Xin Luo, Yihan Luo, Wei Wang

**Affiliations:** 1Business School, Anyang Institute of Technology, West Section of Huanghe Avenue, Anyang 455000, China; 2School of Mathematics and Information Science, Anyang Institute of Technology, West Section of Huanghe Avenue, Anyang 455000, China; 3College of Management, Sichuan Agricultural University, 211 Huimin Rd., Chengdu 130062, China

**Keywords:** public health services, healthy human capital, relative poverty

## Abstract

With the successful completion of the battle against poverty, after 2020, the focus and difficulty of China’s poverty governance will change from solving absolute poverty to alleviating relative poverty. Analyzing and studying the alleviation of relative poverty from the perspective of public health services is in line with the current needs of consolidating and expanding poverty alleviation in China, and it is also of great significance to building a long-term solution mechanism for relative poverty. In this study, basic panel data were constructed by using the data of five CFPS surveys in 2010, 2012, 2014, 2016, and 2018 and matched with the macro data. The correlation between public health services and rural households’ relative poverty was also analyzed by using logit regression analysis and the KHB mediation effect decomposition method. The results show that (1) public health services play a significant role in promoting the accumulation of health human capital, improving individual feasible ability, and alleviating the relative poverty of rural families; (2) the improvement of public health services is conducive to the alleviation of the relative poverty of rural families; (3) we should continue to increase investment in public health care in underdeveloped areas and strive to promote the balanced development of public health services, so as to further consolidate and expand the achievements of poverty eradication.

## 1. Introduction

Since the reform and opening up, China’s economy has developed rapidly, relying on factors such as huge demographic dividends. However, in recent years, China’s population structure has quietly changed, and population aging and other problems are becoming increasingly serious. With the aging of the population and other issues, the medical expenses borne by Chinese residents are also increasing year by year, and the phenomenon of poverty due to illness can occur. At the current stage, the total amount, quality, and structure of China’s public health resources have become increasingly unable to meet the growing needs of the overall population for public health services, and the shortcomings in public services are also one of the reasons for poverty in many places [1]. Poor people and other low-income groups cannot afford high health costs because of their low income. In addition, diseases will further affect their family production capacity, exacerbate poverty, and cause some groups to fall into or return to a state of poverty caused by disease. Continuously improving the development level of public health services is not only the function of the government but also a requirement for sustainable economic development [2]. Constantly promoting the equalization of public health services is also conducive to further narrowing the gap in the availability of services between regions, cities, and villages, so as to ensure that the overall population can enjoy basic public health services. Analyzing and studying the alleviation of relative poverty from the perspective of public health services is in line with the current needs of consolidating and expanding poverty alleviation in China, and it is also of great significance to building a long-term solution mechanism for relative poverty. Solving the problem of relative poverty with public health services is an inevitable requirement to ensure that the overall population shares the fruits of reform and development, which has more important practical value.

Relative poverty is defined from the perspective of “inequality” and “social stratification”. The world bank defined the relative poverty in the 1981 world development report: if some individuals, families and groups cannot obtain the social recognized average living conditions, participation in social activities, and other opportunities, they will be in a state of poverty. Relative poverty itself shows that poverty is relative, which is the most fundamental difference between relative poverty and absolute poverty [3]. Existing research has not reached a consensus on the measurement method of relative poverty, which can be generally classified into four categories. (1) Standard measure of median income level. Townsend emphasized that on the basis of the normal living standard and with reference to the normal social groups, the relatively poor people are those below the average income level [4]. At present, the EU sets the relative poverty line as 60% of the median disposable income per capita, and recommends that 40% and 50% of the median be used as reference indicators. On the basis of reference to EU standards, some Chinese scholars advocated that 40% of the median disposable income per capita should be taken as the relative poverty standard and adjusted according to a certain period [5,6]. (2) Standard measurement method of average income level. The standard measurement method of average level takes a certain proportion of average income as the standard for measuring poverty [7,8]. Some Chinese scholars suggested that 40–50% of the per capita disposable income of residents be taken as the relative poverty standard, and analyzed the relative poverty situation of urban and rural residents [9,10]. (3) Compound number level standard measurement method. Some scholars believe that there is no fundamental difference between relative poverty measured by median or average [11,12]. Chakravarty et al. [13] introduced the Gini coefficient to weight the absolute poverty line and median to construct the relative poverty standard. Ravallion and Chen used the average value after Gini coefficient reduction as the comparison income to measure the global relative poverty [11]. In addition, some scholars used the ratio between the weighted income level of the highest income group of 40% and the weighted average income level of the income group below 60% to measure the degree of social relative poverty [3,14]. (4) Multidimensional relative poverty level measurement method. Sen realized the transformation from the perspective of income to the perspective of ability in the interpretation of poverty, and extended the poverty measurement standard to the multidimensional perspective including the basic ability to obtain food, drinking water, housing, education, health and sanitation [15]. Wang X. and Feng H. pointed out that after 2020, China should adopt the multi-dimensional relative poverty standard according to the actual situation, and put forward an indicator system including three dimensions of economic, social development, and ecological environment [16]. In addition, many Chinese scholars believe that Multidimensional Poverty standards including income, health, education, living conditions, medical and health care, ecological environment, and other dimensions should be established [17,18,19,20,21].

Previous studies on public health services mostly focused on institutional innovation and arrangement and the supply mode of public health services [22,23,24,25,26,27,28]. Public service is an important perspective to study the effect of poverty reduction. Public services include education, health care, social security, and other aspects, and the supply of public services has a significant poverty reduction effect [29]. Providing better guaranteed public services to low-income groups is conducive to helping low-income people get out of poverty [30], and pro-poor growth needs to improve the supply of public services including health care, public assistance, and education [31]. The government has increased investment in basic public services and improved the level of basic public services and equalization, which has played a significant role in reducing the incidence of poverty [32]. Therefore, after effectively identifying the poor, it is necessary to integrate poverty cash assistance and public services to help solve the problem of poverty [33].

Previous studies have shown that public education and public health services have an “enabling” role, which has a greater poverty reduction effect than other public services [1,17,29]. Zhu L. found that measures such as attaching importance to public investment in basic education and health services and providing cheap or free public services to low-income groups in Tibet have reduced economic inequality to a certain extent and narrowed the gap between regions, industries, and groups [34]. The medical security provided by the government can not only directly alleviate poverty but also reduce the incidence of poverty in rural areas by increasing the per capita income of farmers, which has an important impact on rural families [10]. Wang Y. and Liu L. used micro-data to find that the popularization of the new rural cooperative medical system has significantly improved the short-term health human capital level of residents in rural areas, and public medical insurance plays an important role in reducing poverty and increasing income [35]. Wu B. found through research that the poverty reduction effect of the new rural cooperative medical insurance on different poverty standards is different [36]. Public health services and medical insurance have an obvious poverty reduction effect [37]. Some studies also suggest that the medical insurance for urban and rural residents has a more obvious poverty reduction effect than the new rural cooperative medical system [38]. In short, health care is an essential and basic service for poverty eradication, and the government should provide public fund subsidies to help achieve the goal of poverty reduction [39]. To build a governance mechanism for relative poverty, we still need to strengthen social security and other public services. Finance will also play a major role in alleviating relative poverty [17].

To sum up, there is extensive research in the academic community on the effect of poverty reduction, and public health services have become an important perspective for studying the effect of poverty reduction. In general, academic research on relative poverty and public services is becoming refined. These research results also provide a better theoretical guidance for China’s poverty alleviation. However, combing the literature, it is found that scholars pay more attention to the alleviation of absolute poverty, and there is less research on the alleviation of relative poverty by public health services, and these studies are not comprehensive in discussing the poverty reduction mechanism of public health services. In view of this, this paper provides a corresponding supplement to the research at the micro-data level. In this study, basic panel data were constructed by using the data of five CFPS surveys in 2010, 2012, 2014, 2016, and 2018 and matched with the macro-data. The correlation between public health services and relative poverty alleviation was also analyzed by using logit regression analysis and the KHB mediation effect decomposition method. The results show that public health services play a significant role in promoting the accumulation of health human capital, improving individual feasible ability, and alleviating family poverty. The improvement of public health services is conducive to the alleviation of family relative poverty. Public health services directly affect the level of individual feasibility ability and can improve individual self-anti-poverty ability. In addition, public health services and family health investment work together on the accumulation of health human capital within the family, through the accumulation of human capital to promote the improvement of individual or family income level to alleviate the problem of relative poverty. Studying the impact of public health services on family poverty from the family level can lead to more intuitive and detailed analysis results, with a view to providing a corresponding reference for targeted poverty alleviation.

## 2. Analysis Framework and Research Assumptions

Human beings have many basic needs, such as nutrition, safe drinking water, health, housing, and education. The lack of any one of them will have a negative impact on the satisfaction of other basic needs. Low health status not only constitutes the cause of poverty but also poverty itself. Health poverty is also an important part of human poverty. Sen explained the problem of poverty from the perspective of “feasible ability” and believed that compared with income poverty, ability poverty should be more extensively studied [40]. Good health is an important “feasible ability”, and it is also the basic condition for individuals to develop other feasible abilities. For example, good health has a positive impact on children’s learning ability, resulting in a higher school completion rate and a higher average number of years of education. Having good health and investing in health are the prerequisites for people to carry out other social and economic activities. The improvement of health status also means the alleviation of health poverty. At the same time, health drives the satisfaction of other needs, which jointly affect the alleviation of relative poverty in families.

In terms of the types of health investment, there are both personal health investment and public health investment. It is far from enough to rely on individuals to invest in their health, especially for low-income poor groups. Malnutrition, disease, and other low health conditions reduce the income-generating capacity of the poor, and coupled with inadequate medical security, backward public health services, and other issues, the poor can easily fall into the poverty trap and even the intergenerational transmission of poverty. In Western countries, the overall health cost accounts for about 8–9% of GDP, while the education expenditure accounts for 6–7%. The health cost has exceeded the education expenditure. Public health investment in China mainly depends on various public health services provided by the government. Therefore, Hypothesis 1 is proposed:

**Hypothesis** **1** **(H1).**
*Public health services help alleviate the relative poverty of families.*


Early economists studied the impact of educational human capital on economic development but paid less attention to healthy human capital. However, people can provide more effective human capital services only when they are alive and healthy. Mushkin formally included health and education in the two major components of human capital in his article, elaborated on the interactive relationship between them, and studied economic development from the perspective of health human capital investment [41]. Grossman introduced health into the utility function, expanded the health production function on the basis of the Becker family production function, and created the health capital demand theory [42,43]. Since then, many scholars have carried out research on healthy human capital on the basis of Grossman’s theory. Health has increasingly attracted widespread attention, and health and education are generally considered to be two important components of human capital. Healthy human capital is an extension of the initial health state [44], and the human capital stock of workers is mainly composed of health, knowledge, skills, and work experience [2].

The deterioration of health status will reduce the effective working time of individuals, thus reducing the quality of human capital and further leading to the reduction in income. Good health is a necessary condition to ensure that people can provide effective working time. Better health will lead to higher quality human capital accumulation. Wang Y. and Yin Z. used the data of the China Health and Nutrition survey to find that the improvement of nutrition structure can promote the increase in farmers’ income, and healthy human capital has a significant contribution to the increase in farmers’ income in China [45]. According to the estimated results of Gyimah-Brempong and Wilson, 22–30% of the growth rate of per capita income in Sub-Saharan Africa can be attributed to health factors [46]. Similarly, Weil used microeconomic data (such as height and adult survival rate) to establish average health indicators and found that as much as 22.6% of the cross-border differences in per capita income were caused by health factors, which was much the same as the proportion of education in human capital and greater than the share of physical capital [47]. Access to high-level public health services helps improve the health level of individuals and prolong their life expectancy. At the same time, the higher the level of individual health, the more conducive to improving the efficiency of individual work and learning and increased output. The healthier the physical condition, the more conducive it is for individuals to carry out learning and work, improve the feasibility and ability of individuals, and improve the level of knowledge, so as to have a more competitive advantage in the labor market and thus obtain a higher income to relieve poverty. Therefore, Hypothesis 2 is proposed:

**Hypothesis** **2** **(H2).**
*Public health services can help alleviate the problem of relative poverty by increasing the accumulation of family health human capital.*


Based on the above research assumptions, Figure 1 presents the analytical framework of this paper.

## 3. Model Setting, Data Source, and Variable Description

### 3.1. Model Setting

The econometric model constructed in this work analyzes the impact of public health services on the relative poverty of rural families, so the explanatory variable of the econometric model is whether the family is poor. The meaning of the explained variable is that with the established relative poverty line standard, if the annual per capita net income of the family is equal to or lower than the poverty line standard, the family is considered to be in poverty, and the value is assigned as 1; otherwise, the value is 0. Referring to existing studies [48], 40% of the per capita net income of rural households in each province was taken as the relative poverty line standard of each province and used to measure the relative poverty level of families in this study.

The core explanatory variable of the model is public health services, and the local per capita financial public health expenditure is used as the proxy variable. The local per capita financial public health expenditure reflects the size of local investment in health care to a certain extent, and also reflects the importance of local governments to the level of health care. It is generally believed that higher level of investment is conducive to the improvement of local medical and health level and affects the accumulation of local health human capital. Specific to a single family, the medical and health security a family receives is conducive to reducing the burden of family medical and health care and to the accumulation of family health human capital, promoting the improvement of family income level.

Considering that the explained variable is a binary variable, the logit model was used to study the poverty reduction effect of public health services. The equation to be estimated is:(1)$$\mathrm{Logit}\left(\mathrm{Pov}\right)=\ln\frac{{\mathrm{Pov}}_{\mathrm{it}}}{1-{\mathrm{Pov}}_{\mathrm{it}}}=\alpha_{0}+\beta_{1}{\mathrm{Pubmed}}_{\mathrm{it}}+\beta_{2}X+\varepsilon_{\mathrm{it}}$$

In Formula (1), $i\left(i=1,2\ldots n\right)$ represents an individual family, $t\left(t=1,2\ldots T\right)$ represents a period, $\alpha_{0}$ is a constant term, $\beta_{1}\;{\mathrm{and}\;\beta }_{2}$ are the parameter to be estimated, and $\varepsilon_{\mathrm{it}}$ is a random disturbance term. Pov is a binary variable of family poverty status, with poverty status taken as 1 and non-poverty status taken as 0. Pubmed is a public health service variable. In this study, the per capita financial health expenditure was used as a proxy variable.

In order to alleviate the omission and error estimation of the model, a series of control variables X were also introduced that may affect family income and then family poverty, namely a series of family, village, and provincial characteristic variables. Among them, the family characteristic variables include the age of the head of household, the education level of the head of household, the distance from the family to the nearest business street, the annual per capita health care expenditure of the family, whether the family is engaged in agricultural production, and the family burden coefficient. The characteristic variable of the village is the distance between the village and the county. Provincial characteristic variables are the industrial structure characteristics of each province.

Previous studies have shown that there may be a certain correlation between public health services, health human capital, and family relative poverty. This paper attempts to explain the poverty reduction effect of public health services from the perspective of health human capital accumulation. Family health human capital (heapop) was selected as an intermediary variable for analysis. The mediation effect model is:(2)$${\mathrm{Heapop}}_{\mathrm{it}}=b_{0}+b_{1}{\mathrm{Pubmed}}_{\mathrm{it}}+b_{2}X+\mu_{\mathrm{it}}$$
(3)$$\mathrm{Logit}\left(\mathrm{Pov}\right)=\alpha_{0}+\beta_{1}{\mathrm{Pubmed}}_{\mathrm{it}}+\beta_{2}{\mathrm{Heapop}}_{\mathrm{it}}+\beta_{3}X+\varepsilon_{\mathrm{it}}$$

If the following three conditions are met at the same time, it indicates that the intermediary effect of family health human capital exists: first, $b_{1}$ in Formula (2) is significant; that is, there are significant differences in family health human capital with different levels of public services. Second, $\beta_{2}$ in Formula (3) is significant, indicating that health human capital significantly affects family poverty. Third, $\beta_{1}$ from Formula (1) to Formula (3) becomes insignificant, or $\beta_{1}$ is significantly smaller.

### 3.2. Data source and Variable Description

The data used in this paper are the five phase panel data of micro and macro nested matching. Among them, the micro part comes from the data of China Family Panel Studies (CFPS) collected by the China Social Sciences Research Center of Peking University. CFPS data mainly collects and analyzes data at the three levels of individual, family, and community, and conducts corresponding surveys on economic activities, population migration, health, education, and other topics of Chinese families. The data cover 25 provinces across the country, and five rounds of follow-up surveys have been carried out. In terms of micro-data processing, adult data, children’s data, and community data in CFPS data were matched with family data, invalid and missing data were eliminated in the sample, and some midway or newly added samples were deleted. Finally, a total of 3978 continuously tracked rural household samples were retained. The macro part comes from the corresponding year’s China Urban Statistical Yearbook, China Statistical Yearbook and EPS data platform. Then, the macro-variable data matching was nested, and finally, the five phase panel data were formed.

Table 1 shows the variables selected by the regression model and their detailed descriptions and descriptive statistics corresponding to the variables. It can be seen from Table 1 that the mean value of relative poverty of the explained variable families is 0.21072, and the standard deviation is 0.40783, indicating that there are still many families in a state of relative poverty. The standard deviation of per capita financial health expenditure in the core explanatory variable region is 268.54480, indicating that there is a certain degree of difference in health investment between provinces, but this difference is not particularly large. The intermediary variable family health human capital adds the number of adults and children in the family who have self-rated health and then divides it by the total number of families. The definitions and numerical distribution of other control variables are shown in Table 1.

## 4. Analysis of Empirical Results

### 4.1. Basic Regression of Poverty Reduction in Public Health Services

Table 2 shows the basic regression results of poverty reduction in public health services using the time-fixed effect logistic regression method. It can be seen from the results in the table that there is a negative relationship between the per capita financial health care expenditure and the relative poverty of families, and it is highly significant at the significance level of 1%. The regression results show that for every one dollar increase in public health expenditure per capita, the incidence ratio of relative poverty in households will be 0.99791; that is, the incidence ratio of relative poverty will be 0.99791 times that of the original, a decrease of 0.20945% over the previous period. It is further known from the marginal effect that for each unit of increase in public health expenditure per capita, the probability of families falling into relative poverty is reduced by 0.02671%. The improvement of public health service level is conducive to alleviating the problem of relative poverty of families, which verifies Hypothesis 1.

Table 2 also reports the impact of other control variables on the relative poverty of households. The age of the head of household has an inverted “U” effect on family poverty, the age of the head of household has a negative relationship with the relative poverty of the family, and the square of the age of the head of household has a positive relationship with the relative poverty of the family, both of which are highly significant at the significance level of 1%. This is related to the knowledge and experience accumulation of the head of household, which will help reduce the relative poverty of the family when making relevant family decisions at the initial stage. With the increase in age, however, this promoting effect gradually disappears. The older the head of household, the more conservative it may be, which is not conducive to the increase in family income. There is a significant negative relationship between the education level of the head of household and the relative poverty of the family. On the one hand, the education level of the head of household will affect their own earning ability, and on the other hand, it also reflects the accumulation of health knowledge of the head of household. The logarithm of per capita health care expenditure of families has a significant negative relationship with relative poverty. The more families spend on health care expenditure to some extent reflecting their higher income level, the lower the probability of falling into poverty. There is a significant positive relationship between family burden coefficient and family geographical location and relative poverty, which indicates that the heavier the family burden and the more remote the location, the easier it is to fall into relative poverty. There is a significant negative relationship between the proportion of the tertiary industry and the relative poverty of families. The proportion of the tertiary industry has a negative relationship with the relative poverty of the family, but it is not significant.

### 4.2. Mediating Effect of Family Health Human Capital

Taking family health human capital as the intermediary variable for analysis, the model (1) in Table 3 first tests the differences of family health human capital with different public service levels. According to the regression results in the table, the positive impact of per capita financial public health expenditure on family health human capital is significant at the level of 1%, and the impact coefficient is 0.00007, indicating that for each unit of per capita financial public health expenditure, family health human capital increases by 0.00007 units, and the increase in per capita financial public health expenditure is conducive to the accumulation of family health human capital. Model (2) is the regression result after adding family health human capital variables. It can be seen from the table that for every unit of family health human capital increases, the probability of a family falling into relative poverty decreases by 13.23%, and family health human capital has a significant impact on family relative poverty. The per capita financial medical and health expenditure still has a significant impact on the relative poverty of families, but compared with the results in Table 2, it is found that when the per capita financial medical and health expenditure is increased by one unit, the probability of families falling into relative poverty is reduced by 0.02593%, which is smaller than the previous reduction of 0.02671%. This shows that the results meet the assumptions of some intermediary variables, and family health human capital is established as part of the intermediary variables, which can partially explain the impact of public health services on family relative poverty and verifies Hypothesis 2.

In order to further verify the mediating effect of family health human capital, this study used the KHB mediation effect decomposition method to decompose the relevant impact effects, and the results are shown in Table 4. It can be seen from the table that the increase in public health services will reduce the probability of families falling into relative poverty by 0.00028. With the control of family health human capital, the impact of public health services becomes 0.00023, leaving an indirect impact of 0.00005. The total effect is 1.2279 times the direct effect, and 18.56% of the total effect of public health services on poverty reduction comes from family health human capital. On the whole, family health human capital passed the intermediary effect test, which is consistent with the test results in Table 3.

### 4.3. Robustness Test

In order to ensure the robustness and reliability of the model estimation results, a series of robustness tests were carried out in this study. The specific results are shown in Table 5. Commonly used robustness testing methods include the replacement model estimation method, replacement of explained variables or replacement of core explanatory variables. In model (1) in Table 5, the probit regression method is shown. In model (2), the explanatory variable is replaced with POV1, which means that 50% of the per capita net income of rural households in each province in the current year is used as the relative poverty line standard of each province. When the per capita net income of households is equal to or lower than the poverty line standard, it is assigned as 1; otherwise, the value is 0. In model (3), POV1 is used as the explained variable, and probit estimation is used in the estimation method. The three models are all time-fixed effect models. The estimation results of the three models show that after controlling the corresponding family, village/residence, and provincial characteristic variables, the symbols and significance of the core explanatory variables and intermediary variables change little. After replacing the explained variables and the estimation methods, the total, direct, and indirect effect coefficients of public health services on the relative poverty of families are negative, the coefficient changes slightly, and all pass the significance test. This further proves the credibility of the results of this study, indicating that family health human capital can be used as an intermediary variable to explain the impact of public health services on family poverty.

## 5. Discussion

This paper mainly focuses on the theme of public health services and relative poverty alleviation, makes a corresponding theoretical analysis of the mechanism of public health services to reduce poverty, and presents an econometric model for empirical research on the relationship between the two. Public health services have a positive impact on improving the health of residents. A higher level of public services will bring lower mortality and higher life expectancy [2,30,49,50]. The government’s popularization of individual health knowledge can promote the improvement of residents’ health awareness, and the emphasis on vaccination and disease prevention can enhance the ability of individual residents to prevent and control diseases and improve the health level of residents [51]. The per capita income and public health expenditure have positive impact on health status, and the poverty rate decreased with the improvement of health status [52]. The poverty reduction effect of government health expenditure is significant [22,34,53].

The conclusion of this paper is in good agreement with existing research. Different from the existing literature, this paper not only tests the direct effect of public health services on poverty reduction, but also empirically tests its indirect effect. The empirical results prove the two hypotheses proposed in this paper and pass the robustness test. The contributions of this study are: (a) using multi-period panel data matched with micro-survey data and macro-data, this paper empirically analyzes the impact of public health services on Rural Households’ relative poverty. (b) This paper discusses the mechanism of public medical and health services on the relative poverty of rural families. (c) It verifies the intermediary role of health human capital in the poverty reduction of public health services, and provides a theoretical basis for the later construction of a long-term solution mechanism for relative poverty.

Compared with existing research, this paper has some innovations in research content. However, it is undeniable that this work still has some deficiencies in theoretical analysis and empirical analysis. Limited by the data, this study only used the per capita financial expenditure on public health services as its proxy variable when calculating public health services. When calculating the indicators of relative poverty, only the indicators at the income level were considered. In reality, relative poverty is not only reflected in income but also in other material and spiritual aspects. Public medical and health services cover many contents, and the differences between regions are poor. Due to the limitations of research space and data, this study did not classify and analyze public health services. Corresponding field research activities are also planned in the future, which will be more combined with poverty alleviation policies and activities. These will be the direction of further research in the future.

## 6. Conclusions

This paper first analyzed the background and research status of public health services and poverty alleviation, then reviewed and summarized the theoretical research on public health services and poverty alleviation, and analyzed the internal mechanism of public health services to reduce poverty. After analyzing the theoretical mechanism, this paper established an econometric model of public health service poverty reduction, using which the corresponding variables and data were selected to carry out empirical analysis. Basic regression was first carried out on the model, then intermediary variables were introduced, and the KHB intermediary effect decomposition method was used to decompose and analyze the total effect. The substitution estimation method and the substitution of explained variables were then adopted to test the robustness of the model. The estimated results show that the poverty reduction effect of public health services is relatively significant. The estimation results show that the improvement of the level of public health care contributes to the alleviation of the problem of family relative poverty. The human capital of family health is established as part of the intermediary variable, which can partially explain the impact of public health services on family relative poverty. This conclusion is robust and reliable.

Health and education are the two main factors in the accumulation of human capital. Good health can not only promote the accumulation of healthy human capital but also help prolong the life of individuals and improve their feasibility. Poor groups, due to the restriction of income level, are unable to invest substantially in health care. Families suffering from the health impact of diseases will have their income sharply reduced due to the lack of labor, and the high medical costs will aggravate the family’s poverty. Over time, it is very easy to enter a vicious circle of a poverty trap. Public health services play a significant role in promoting the accumulation of health human capital, improving the efficiency of individual work and learning, and alleviating family poverty. Public health investment and family health investment work together in the healthy accumulation of human capital within the family through the accumulation of human capital to promote economic growth and then promote the improvement of individual or family income level to alleviate the problem of relative poverty. In addition to the indirect poverty reduction effect of health human capital, public health services also directly affect the level of individual feasibility ability so as to make the individual labor market more competitive in order to obtain higher income to relieve poverty. Therefore, the government needs continue to implement the China health poverty alleviation project. The strategy is to provide medical insurance, medical assistance, and other measures to reduce the probability of poverty caused by illness in poor families. At the same time, we should increase investment in, further the reform of, and improve the public health care system. We should improve public health service projects, ensure that medical and health services can benefit the overall population, and effectively ensure that the overall population shares the fruits of reform, opening up, and development.

## Figures and Tables

**Figure 1 ijerph-19-11089-f001:**
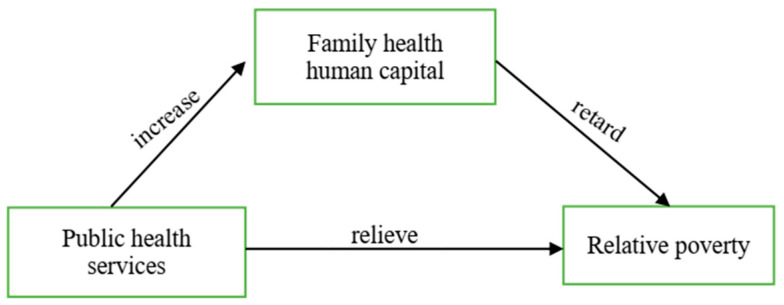
Analysis framework of poverty reduction by public health services.

**Table 1 ijerph-19-11089-t001:** Variable description and descriptive statistics.

Variable Name	Variable Description	Mean	SD ^a^	MIN	MAX
Family relative poverty (Pov)	1 = Poverty 0 = Non-poverty	0.21072	0.40783	0	1
Public health services (Pubmed)	Per capita financial medical and health expenditure (CNY)	637.08610	268.54480	199.83100	1704.61700
Age of householder (Age)	Year	47.65101	16.75277	10	99
Education level of householder (Aedu)	Year	5.89187	4.40064	0	22
Family health human capital (Heapop)	Family self-rated healthy population/total family population	0.81935	0.26627	0	1
Time spent by the family away from the nearest business street (Cdis)	minute	36.60226	59.45423	1	1440
Whether the family is farming (Farm)	1 = Engage in agricultural work, 0 = not engaged in agricultural work	0.81585	0.38762	0	1
Family health expenditure (Afmedcost)	Per capita health care expenditure of households (CNY)	1273.3460	4303.7340	0	246,666.70
Family burden coefficient (Bur)	Family non-working age population/working age population	0.36143	0.59995	0	10
Geographical location of the village (Cdistance)	Distance between village and county (LI)	44.11037	37.13605	0	280
Industrial structure (Third)	Proportion of tertiary industry in the province (%)	41.90452	6.79345	29.30	69.17880

^a^ SD = standard deviation.

**Table 2 ijerph-19-11089-t002:** Logit regression of poverty reduction by public health services.

Independent Variable	Coefficient	Odds Ratio	dy/dx
Pubmed	−0.00210 *** (0.00026)	0.99791 *** (0.00026)	−0.00027 *** (0.00003)
Age	−0.04382 *** (0.00708)	0.95713 *** (0.00678)	−0.00558 *** (0.00090)
Age2	0.00063 *** (0.00007)	1.00063 *** (0.00007)	0.00008 *** (0.00001)
Aedu	−0.05400 *** (0.00627)	0.94743*** (0.00594)	−0.00688 *** (0.00080)
Lnafmedcost	−0.02146 *** (0.00553)	0.97877 *** (0.00541)	−0.00273 *** (0.00070)
Cdis	0.00214 *** (0.00045)	1.00214 *** (0.00045)	0.00027 *** (0.00006)
Farm	−0.27129 *** (0.06154)	0.76240 *** (0.04692)	−0.03456 *** (0.00781)
Bur	0.23207 *** (0.03927)	1.26121 *** (0.04953)	0.02956 *** (0.00499)
Cdistance	0.00392 *** (0.00076)	1.00393 *** (0.00076)	0.00050 *** (0.00010)
Third	−0.00835 (0.00553)	0.99168 (0.00548)	−0.00317 (0.00070)
Constant term	−0.34084 (0.29349)	0.71117 (0.20872)	_
Observations	19,890	19,890	19,890

Note: The standard errors are presented in brackets. *** is statistically significant at 1%.

**Table 3 ijerph-19-11089-t003:** Intermediary effect of family health human capital.

Independent Variable	(1)	(2)	
	Heapop	Pov	
	Coefficient	Odds ratio	dy/dx
Pubmed	0.00007 *** (0.00002)	0.99797 *** (0.00026)	−0.00026 *** (0.00003)
Heapop		0.35703 *** (0.03058)	−0.13132 *** (0.01088)
Lnafmedcost	−0.00722 *** (0.00043)	0.96972 *** (0.00537)	−0.00392 *** (0.00071)
Age	0.00030 (0.00053)	0.95569 *** (0.00673)	−0.00578 *** (0.00090)
Age2	−0.00002 *** (0.00001)	1.00062 *** (0.00007)	0.00008 *** (0.00001)
Cdis	−0.00008 * (0.00005)	1.00206 *** (0.00044)	0.00026 *** (0.00006)
Cdistance	−0.00008 (0.00007)	1.00379 *** (0.00075)	0.00048 *** (0.00009)
Farm	0.02003 *** (0.00486)	0.78939 *** (0.04845)	−0.03015 *** (0.00780)
Bur	0.05296 *** (0.00310)	1.34931 *** (0.05289)	0.03820 *** (0.00499)
Aedu	0.00412 *** (0.00048)	0.95211 *** (0.00594)	−0.00626 *** (0.00080)
Third	−0.00095 ** (0.00045)	0.99091 * (0.00542)	−0.00116 *** (0.00070)
Constant term	0.86184 *** (0.02320)	1.78025 * (0.53466)	—
Observations	19,890	19,890	

Note: The standard errors are presented in brackets. ***, **, and * are statistically significant at 1%, 5%, and 10%, respectively.

**Table 4 ijerph-19-11089-t004:** Breakdown of the impact of public health services on poverty reduction.

Core Explanatory Variables	Intermediary Variable	Dependent Variable (POV)	
Pubmed	Heapop	Total effect	−0.00028 **
			(0.00012)
		Direct effect	−0.00023 *
			(0.00012)
		Indirect effect	−0.00005 ***
			(0.00001)
	Indirect effect/Total effect		18.56%

Note: The standard errors are presented in brackets. ***, **, and * are statistically significant at 1%, 5%, and 10%, respectively.

**Table 5 ijerph-19-11089-t005:** Robustness test of public health services for poverty reduction.

Estimation Method	Probit	Logit	Probit
Dependent Variable	Pov	Pov1	Pov1
Model Number	(1)	(2)	(3)
Pubmed	−0.00025 *** (0.00003)	−0.00030 *** (0.00004)	−0.00028 *** (0.00004)
Heapop	−0.13364 *** (0.01104)	−0.16094 *** (0.01215)	−0.16899 *** (0.01231)
Other control variables	Yes	Yes	Yes
Observations	19,890	19,890	19,890
Total effect	−0.00017 ** (0.00007)	−0.00024 ** (0.00011)	−0.00014 ** (0.00007)
Direct effect	−0.00014 ** (0.00007)	−0.00020 * (0.00011)	−0.00011 * (0.00007)
Indirect effect	−0.00003 *** (0.00001)	−0.00004 *** (0.00001)	−0.00003 *** (0.00001)
Indirect effect/Total effect	17.56%	16.58%	23.55%

Note: The regression results reported by the three models are the average marginal effect coefficient. The standard errors are presented in brackets. ***, **, and * are statistically significant at 1%, 5%, and 10%, respectively.

## Data Availability

The data that support the findings of this study are available from the authors upon reasonable request.

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
