# Peer review of "Public Health Services, Health Human Capital, and Relative Poverty of Rural Families"

_ijerph, 2022, doi:10.3390/ijerph191711089_

Round 1

Reviewer 1 Report

This article analyses and discusses the correlation between public health services and rural households' relative poverty using a huge amount of data from the China Family Panel Studies (CFPS) database and logit regression analysis and the KHB mediation effect decomposition method. For all the above, the research does not present substantive problems, although some formal aspects can be mentioned that must necessarily be corrected.

(1)  The concept of Relative Poverty needs to be clearly defined.

(2)  It is suggested that the variables of Public health services(Pubmed)and Family health expenditure(Afmedcost)should be logarithmic.

(3)  As for the test of mediating effect of family health human capital, it is suggested to refer to Wen Zonglin's mediating effect test model.

Author Response

Dear reviewer and editor,

Thank you very much for your valuable advice. Your opinions are very important. We have carefully revised the following according to the review opinions of the paper.

Point1: The concept of Relative Poverty needs to be clearly defined.

Thank you for your valuable advice. According to your suggestion,we have explained the definition of relative poverty in the literature review.

Point 2:It is suggested that the variables of Public health services(Pubmed)and Family health expenditure(Afmedcost)should be logarithmic.

Thank you for your valuable advice. According to your suggestion ,the variable of household health expenditure has taken logarithm in the regression.

Point 3: As for the test of mediating effect of family health human capital, it is suggested to refer to Wen Zonglin's mediating effect test model.

Thank you for your valuable advice. The dependent variable in this paper is a binary variable. The Monte Carlo studies proposed by Karlson, Holm and Breen (2010) and Karlson and Holm (2011) show that the KHB method proposed by Karlson, Holm and Breen can decompose the influence of discrete variables and continuous variables, can be expanded to adapt to the influence of average bias, provides statistical tests for analysis and derivation, and is simple and intuitive. In fact, the KHB method extends the decomposability of linear models to nonlinear probabilistic models. KHB has unique advantages in the intermediary analysis test of nonlinear regression models.

Reviewer 2 Report

Public Health Services, Health Human Capital, and Relative 2 Poverty of Rural Families cover an important topic. I believe this paper can add to the study of public health. My main suggestion is to define the variables used in the study. The paper also introduces various concepts without providing clear definitions. The authors should also highlight the paper’s unique contribution to the field.

Introduction

In the introduction, the authors mention public medical and health services. Do they mean public health and medical services or government-provided health and medical services? Clarifying and expanding the term would be helpful to increase the readability. In addition, including a brief definition of relative and absolute poverty in the introductory section would be helpful.

Literature Review

The authors should provide a clear definition of relative poverty and highlight its difference from absolute poverty.

Also, the authors mention the notion of health human capital without clearly articulating how they define the term.

Analysis framework and research assumptions

 The analytical framework introduces several different concepts without providing a clear definition. The section would be strengthened by providing a clear definition of feasible ability. This section also requires a clear definition of human capital.

Method setting, data source, and variable description

In lines, 201-206, the authors explain how they measure relative poverty, which is “40% of the per capita net income of rural households.” The authors mention relative household poverty as one of their variables. Lines 208 and 209 introduce another variable, per capita financial health expenditure. I wonder how the authors justify using two different units of analysis where one variable focuses on a household while the other focuses on an individual unit. Is this one of the limitations of the study? If so, the authors need to address it.

The second section explains data sources. It might help to begin the method section by discussing the data sources.

Analysis of empirical results

The finding section needs to expand a little on the reported statistical results and what it means for the study.

 Discussion

The discussion section should focus more on outlining the paper’s unique contribution. Currently, the focus is more on the study’s limitation than its contribution. Also, some connections with past studies would be helpful here.

Conclusion and policy recommendations

On line 454, what do the authors mean by “healthy poverty alleviation”? Is a reduction in poverty associated with health?

Author Response

Dear reviewer and editor,

Thank you very much for your valuable advice. Your opinions are very important. We have carefully revised the following according to the review opinions of the paper. We have revised the normative of references and references.

Point 1: In the introduction, the authors mention public medical and health services. Do they mean public health and medical services or government-provided health and medical services? Clarifying and expanding the term would be helpful to increase the readability. In addition, including a brief definition of relative and absolute poverty in the introductory section would be helpful.

Thank you for your valuable advice. The core explanatory variable of this study is public health service. We have revised it in the introduction. The definition of relative poverty is explained in the literature review.

Point 2: The authors should provide a clear definition of relative poverty and highlight its difference from absolute poverty. Also, the authors mention the notion of health human capital without clearly articulating how they define the term.

Thank you for your valuable advice. We have explained the relative poverty and healthy human capital according to your suggestion.

Point 3:  The analytical framework introduces several different concepts without providing a clear definition. The section would be strengthened by providing a clear definition of feasible ability. This section also requires a clear definition of human capital.

Thank you for your valuable advice. We have explained the relative poverty and healthy human capital according to your suggestion.

Point 4: In lines, 201-206, the authors explain how they measure relative poverty, which is “40% of the per capita net income of rural households.” The authors mention relative household poverty as one of their variables. Lines 208 and 209 introduce another variable, per capita financial health expenditure. I wonder how the authors justify using two different units of analysis where one variable focuses on a household while the other focuses on an individual unit. Is this one of the limitations of the study? If so, the authors need to address it. 

The second section explains data sources. It might help to begin the method section by discussing the data sources.

Thank you for your valuable advice. Indeed, as you said, the research units of the dependent variable and the core explained variable are different, which may cause some confusion to the readers. If the public health service accessibility data at the household level can be obtained, it will greatly improve the accuracy of this study. The empirical analysis serves for the theoretical analysis. In the previous theoretical analysis, this paper analyzes in detail the role mechanism of public health services on relative poverty, and this role mechanism has its rationality. When reading the existing literature, we found that some authors matched and nested macro data and micro data in their research. In order to solve the problems as much as possible, this paper controls the relevant factors at the family, village and provincial levels. And in the empirical analysis, the intermediary effect is analyzed. The intermediary variable is the health human capital stock at the family level. The idea of empirical design is close to the previous theoretical analysis. Therefore, this paper adopts this empirical method in the subsequent analysis.

Point 5: The discussion section should focus more on outlining the paper’s unique contribution. Currently, the focus is more on the study’s limitation than its contribution. Also, some connections with past studies would be helpful here.

Thank you for your valuable advice. We have explained the relative poverty and healthy human capital according to your suggestion.

Point 6: On line 454, what do the authors mean by “healthy poverty alleviation”? Is a reduction in poverty associated with health?

Thank you for your valuable advice. We are sorry that our expression may have caused you a misunderstanding. We have made changes in the text. The purpose of the China health poverty alleviation project is to promote the medical and health care undertakings at the grass-roots level and in the rural poor areas in China by carrying out a series of public welfare service projects, narrow the health gap between urban and rural residents, improve the quality of life of vulnerable groups, highlight the social fairness and justice of "everyone is healthy", and gradually eliminate poverty caused by illness and poverty caused by illness.

Reviewer 3 Report

A major issue with the paper is that the authors seem to be only concerned with the statistics and made little effort to explain the importance of the study in connection with other studies published in English. The Introduction has only one reference. Except for this journal, almost all publications listed in References are shown in English but most of them are actually published in Chinese journals, so please show the original Chinese title of journals and books. In addition, all of the names mentioned in literature review are Chinese, and very few studies in English are included. I have to wonder, as the paper was submitted to an English journal, why didn’t the authors cite more English studies? Relatedly, it is acceptable for the Conclusion to have no references, but it is not OK for the Discussion to have no references at all. In the discussion section, the authors should have connected the findings of this study to other studies, therefore should have many references in English. This is not simply an issue of references; it reflects the authors’ ignorance of the English literature and lack of engagement with existing research in the West.

Another problem is that key concepts such as ‘relative poverty’ were not explained – how is it defined in China? What is the official threshold of relative poverty in China? And there should be more contextual information, such as how many people are in relative poverty, etc.. Similarly, what is ‘public management theory’? There are no references about this concept. ‘The research on poverty mostly focuses on the solution of absolute poverty and discusses the corresponding solutions to absolute poverty.’ (lines 57-58) Whose research is this? About which countries? What are the references for such claim?

The authors used the micro data collected from the China Household Tracking Survey (CFPS). First of all, you should have spelled out CFPS when it is mentioned for the first time. Second, it has been translated as China Family Panel Studies, not China Household Tracking Survey; otherwise, the shorthand should be CHTS. It covers both rural and urban areas, so the authors should have made it clear how the rural cases were extracted. More importantly, data from five waves (2010 to 2018) were analysed, but no longitudinal changes over these eight years of the key variables were shown. The two hypotheses and the mediation model do not mention any possible changes over time, and the regression equations do not incorporate time, so what is the point of analysing longitudinal data if you don’t plan to show longitudinal changes?

Last paragraph of Introduction summarizes the findings, which should be in the Abstract. The authors should have used the space to explain why this study is needed.

Chinese terms used are not explained, such as "Three Guarantees", and the paper is full of statements that sound like China’s official propaganda.

Lines 107 – 108: sentence was disconnected.

Author Response

Dear reviewer and editor,

Thank you very much for your valuable advice. Your opinions are very important. We have carefully revised the following according to the review opinions of the paper.

Point1: A major issue with the paper is that the authors seem to be only concerned with the statistics and made little effort to explain the importance of the study in connection with other studies published in English. The Introduction has only one reference. Except for this journal, almost all publications listed in References are shown in English but most of them are actually published in Chinese journals, so please show the original Chinese title of journals and books. In addition, all of the names mentioned in literature review are Chinese, and very few studies in English are included. I have to wonder, as the paper was submitted to an English journal, why didn’t the authors cite more English studies? Relatedly, it is acceptable for the Conclusion to have no references, but it is not OK for the Discussion to have no references at all. In the discussion section, the authors should have connected the findings of this study to other studies, therefore should have many references in English. This is not simply an issue of references; it reflects the authors’ ignorance of the English literature and lack of engagement with existing research in the West.

Thank you for your valuable advice. Your opinion is very pertinent and we have benefited a lot. According to your opinions, we have carefully revised the introduction, literature review and discussion in the article. Before that, it was indeed our lack of experience in writing English papers that led to such problems. As the research is about the relative poverty in China, some excellent papers published by Chinese scholars in Chinese journals and English journals are also cited. After revision, the current English references account for more than half. Once again, I would like to express my sincere feelings to you. Thank you for your generous advice.

Point2: Another problem is that key concepts such as ‘relative poverty’ were not explained – how is it defined in China? What is the official threshold of relative poverty in China? And there should be more contextual information, such as how many people are in relative poverty, etc.. Similarly, what is ‘public management theory’? There are no references about this concept. ‘The research on poverty mostly focuses on the solution of absolute poverty and discusses the corresponding solutions to absolute poverty.’ (lines 57-58) Whose research is this? About which countries? What are the references for such claim?

Thank you for your valuable advice. At present, there is no unified definition of the relative poverty standard. On the basis of considering your suggestions, this paper introduces several relatively uniform relative poverty measurement methods in detail in the literature review section. Other contents have also been modified according to your opinions.

Point3: The authors used the micro data collected from the China Household Tracking Survey (CFPS). First of all, you should have spelled out CFPS when it is mentioned for the first time. Second, it has been translated as China Family Panel Studies, not China Household Tracking Survey; otherwise, the shorthand should be CHTS. It covers both rural and urban areas, so the authors should have made it clear how the rural cases were extracted. More importantly, data from five waves (2010 to 2018) were analysed, but no longitudinal changes over these eight years of the key variables were shown. The two hypotheses and the mediation model do not mention any possible changes over time, and the regression equations do not incorporate time, so what is the point of analysing longitudinal data if you don’t plan to show longitudinal changes?

Thank you for your valuable advice. "Panel data" is a data table construction form that can contain more information than "cross-sectional data" and "time series data". The biggest advantage of using panel data is that it can expand the sample size as much as possible and improve the reliability of conclusions. Common panel models mainly include individual fixed effect model, time fixed effect model and two-way fixed effect model. Since there are control variables that do not change with time in our econometric model, the use of individual fixed effects cannot be used in previous estimates. After seeing your suggestion, we reviewed the relevant literature and found that the time fixed effect can be used. The empirical part has used the time fixed effect and revised the results. Thank you very much for your generous advice.

Point4: Last paragraph of Introduction summarizes the findings, which should be in the Abstract. The authors should have used the space to explain why this study is needed.

Chinese terms used are not explained, such as "Three Guarantees", and the paper is full of statements that sound like China’s official propaganda.

Lines 107 – 108: sentence was disconnected.

Thank you for your valuable advice. According to your opinions and the writing requirements of the journal, we have made corresponding amendments

Round 2

Reviewer 3 Report

The issues I pointed out have not been addressed properly, and they appear to be unrepairable. 

Author Response

To reviewer 3:

Dear reviewer and editor,

Thank you very much for your valuable advice. Your opinions are very important. We have carefully revised the following according to the review opinions of the paper.

Point 1 : The issues I pointed out have not been addressed properly, and they appear to be unrepairable. 

Thank you for your valuable advice. Your opinion is very pertinent and we have benefited a lot. According to your opinions, we have carefully revised the introduction. Other contents have also been modified according to your opinions. Before that, it was indeed our lack of experience in writing English papers that led to such problem. Thank you again.